# CNV-Net: Segmentation, Classification and Activity Score Measurement of Choroidal Neovascularization (CNV) Using Optical Coherence Tomography Angiography (OCTA)

**DOI:** 10.3390/diagnostics13071309

**Published:** 2023-03-31

**Authors:** Mahsa Vali, Behzad Nazari, Saeed Sadri, Elias Khalili Pour, Hamid Riazi-Esfahani, Hooshang Faghihi, Nazanin Ebrahimiadib, Momeneh Azizkhani, Will Innes, David H. Steel, Anya Hurlbert, Jenny C. A. Read, Rahele Kafieh

**Affiliations:** 1Department of Electrical and Computer Engineering, Isfahan University of Technology, Isfahan 84156-83111, Iran; 2Retina Ward, Farabi Eye Hospital, Tehran University of Medical Sciences, Tehran 14176-13151, Iran; 3Royal Victoria Infirmary Eye Department, Newcastle Upon Tyne Hospitals NHS foundation Trust, Newcastle upon Tyne NE1 4LP, UK; 4Sunderland Eye Infirmary, Sunderland SR2 9HP, UK; 5Center for Transformative Neuroscience and Institute of Biosciences, Newcastle University, Newcastle upon Tyne NE2 4HH, UK; 6Medical Image and Signal Processing Research Center, School of Advanced Technologies in Medicine, Isfahan University of Medical Sciences, Isfahan 81746-73461, Iran; 7Department of Engineering, Durham University, South Road, Durham DH1 3LE, UK

**Keywords:** optical coherence tomography angiography, choroidal neovascularisation, segmentation, classification, activity score measurement

## Abstract

This paper aims to present an artificial intelligence-based algorithm for the automated segmentation of Choroidal Neovascularization (CNV) areas and to identify the presence or absence of CNV activity criteria (branching, peripheral arcade, dark halo, shape, loop and anastomoses) in OCTA images. Methods: This retrospective and cross-sectional study includes 130 OCTA images from 101 patients with treatment-naïve CNV. At baseline, OCTA volumes of 6 × 6 mm^2^ were obtained to develop an AI-based algorithm to evaluate the CNV activity based on five activity criteria, including tiny branching vessels, anastomoses and loops, peripheral arcades, and perilesional hypointense halos. The proposed algorithm comprises two steps. The first block includes the pre-processing and segmentation of CNVs in OCTA images using a modified U-Net network. The second block consists of five binary classification networks, each implemented with various models from scratch, and using transfer learning from pre-trained networks. Results: The proposed segmentation network yielded an averaged Dice coefficient of 0.86. The individual classifiers corresponding to the five activity criteria (branch, peripheral arcade, dark halo, shape, loop, and anastomoses) showed accuracies of 0.84, 0.81, 0.86, 0.85, and 0.82, respectively. The AI-based algorithm potentially allows the reliable detection and segmentation of CNV from OCTA alone, without the need for imaging with contrast agents. The evaluation of the activity criteria in CNV lesions obtains acceptable results, and this algorithm could enable the objective, repeatable assessment of CNV features.

## 1. Introduction

Age-related macular degeneration (AMD) is the leading cause of sight impairment in developed countries in individuals over 50 years of age [1]. AMD-related visual loss is mostly brought on by choroidal neovascularisation (CNV), an advanced form of AMD [2]. Its primary characteristic is the development of aberrant blood vessels, which originate in the choroid and mostly expand between the Bruch’s membrane and the retinal pigment epithelium (RPE) (Type I) or in the subretinal region (Type II) [3]. 

In order to diagnose and categorise CNV lesions, fluorescein angiography (FA) has long been the gold standard imaging technique, with indocyanine green angiography (ICGA) used to identify their complete extent [4]. However, these techniques are inconvenient for patients, requiring an intravenous injection that can produce symptoms such as nausea, as well as temporarily discolouring tissue, and can potentially produce allergic reactions. They also incur costs due to nurse time and the need for the patient to remain in clinic after the injection. Furthermore, the precise assessment of the vascular features of these neovascularisations is not possible with these dye-based imaging techniques [5]. This is problematic because, clinically, there is a pressing need to enhance the decision-making process for anti-VEGF treatment initiation and the re-treatment of CNV based on the response [6]. 

Recently, optical coherence tomography (OCT) has revolutionised the diagnosis of CNV development with its non-invasive high-resolution scanning capabilities [7]. OCT makes it possible to determine the morphological characteristics of the fibrovascular complex and the extent of intraretinal, as well as subretinal, fluids. These features are routinely used for activity estimation and treatment decisions for different types of CNV. However, the vascular structure is imaged even more poorly with OCT than with FA/ICGA, meaning that dye-based imaging continues to be necessary [8]. 

More recently, the advent of OCT angiography (OCTA) has made it possible to accurately evaluate the microvascular morphology of these lesions with OCT [9,10]. This raises the possibility that CNV could be detected, assessed, and monitored entirely non-invasively, without the need for dyes. This would improve the patient’s experience and safety, as well as reducing costs. However, exclusively using OCTA to identify neovascular activity would require defined OCTA parameters for determining the disease stage of a CNV, including quantitative criteria with proven sensitivity and specificity [5,9,10,11,12,13,14,15]. 

Cascos et al. [9] proposed criteria to determine CNV activity based on OCTA images (Figure 1). A lesion was assessed as an active CNV if it revealed at least three of the following five features:Shape: a well-defined (lacy-wheel or sea-fan shaped) CNV lesion, in contrast to one with long filamentous linear vessels.Branching: numerous tiny capillaries, in contrast to rare large mature vessels.The presence of anastomoses and loops.Morphology of the vessel termini: the presence of a peripheral arcade, in contrast to a “dead tree” appearance.Presence of a perilesional hypointense halo, defined as regions of choriocapillaris alteration, either due to flow impairment, steal, or localized atrophy.

They showed an acceptable agreement between conventional multimodal imaging (CNV detected on at least two of FA, ICGA, and OCT) and the presence of the mentioned CNV activity criteria (on OCTA alone) for the determination of CNV activity. Later, they showed that the combination of branching, anastomoses, the type of vessel termini, and the presence of a hypointense perilesional halo together on OCTA predicted the exudative status of the CNV lesion in 97.6% of cases. Interestingly, they illustrated that the probability of detecting active CNV considering only tiny branching vessels and the peripheral arcade on OCTA (features 2 and 3 above) was 71.23%, which was inferior to a model containing all five criteria [5]. Therefore, it seems that consideration of all the above criteria is necessary to predict the CNV activity status accurately. Although the intergrader correlation coefficient for determining the presence or absence of the mentioned features was high for retina specialists, providing an automatic activity grading for non-retina specialists or retina specialists who are not familiar with these activity criteria would be useful [6]. One of the most popular target diseases for deep learning (DL), a subfield of artificial intelligence (AI), in the field of medical imaging is AMD [16]. 

In this study, we aimed to develop a DL-based decision support based on the mentioned five CNV activity features for ophthalmologists to determine the CNV activity in daily clinical work. To validate the robustness, the algorithms were tested via cross-validation and benchmarked against two retina specialists.

## 2. Materials and Methods

### 2.1. Dataset

This retrospective and cross-sectional study included 130 OCTA images from 101 patients with treatment-naive type 1 or 2 CNVs secondary to AMD who were referred to Farabi Eye Hospital at Tehran University of Medical Sciences in Iran between March 2019 and October 2021. In the corresponding OCT images of the macular area, 101 of the 130 evaluated OCTA images revealed subretinal fluid (SRF), 67 had intraretinal fluid (IRF), 46 had both SRF and IRF, and 13 had neither SRF nor IRF. 

The Declaration of Helsinki’s guiding principles were followed throughout the course of the research, and the protocol for the study was approved by the Administrative Evaluation Board at Tehran University of Medical Science (IR.TUMS.FARABIH.REC.1400.055). All of the patients gave their full informed consent. The exclusion criteria included type 3 CNV, a refractive error greater than 6 diopters or an axial length greater than 26.5 mm, and poor-quality images (scan image quality index 5/10 or worse) due to significant motion, flashing artefacts, shading, or projection artefacts.

At baseline, OCTA volumes of 6 × 6 mm^2^ were obtained using the Optovue RTVue XR Avanti spectral-domain OCT device (Optovue, Inc., Fremont, CA, USA) with the split-spectrum amplitude-decorrelation angiography algorithm. The device produced two OCT volumes consisting of 304 × 304 A-scans each in approximately 2.6 s at a rate of 70,000 A-scans per second. The presence of a neovascular complex was assessed on the outer retina and choriocapillaris en-face images generated by automatic segmentation obtained from the OCTA built-in software. To better visualise the CNV complex, the boundaries of the en-face images were manually adjusted to cover the entire lesion.

We propose a two-stage algorithm for segmenting the vicinity of the CNV into 3 regions (the entire CNV, plus the dark halo and the peripheral arcade) and assigning the activity criteria to each lesion. This was accomplished by using en-face images of the outer retina and choriocapillaris. The area containing the CNV in the outer retinal en-face images (CNV mask) was determined using a Deep-learning (DL)-based method during the segmentation stage. By multiplying the CNV mask in the original image (CNV mask), the background is suppressed while maintaining the background of the CNV vascular area, which is required for the feature extraction in the next step. Furthermore, the dark halo mask and peripheral arcade mask are segmented from the outer retinal and choriocapillaris en-face images, respectively, using classical image processing techniques. These image-patches picked out by three masks are then fed into a DL-based classification stage, where the five activity criteria are automatically assigned to each CNV lesion. Individual binary DL classification models are used to determine the presence or absence of each activity criterion. Due to the scarcity of data, the transfer learning (TL) technique with pre-trained models is used. The network is then tailored by re-training a number of network parameters with the available CNV images. The proposed CNV-Net algorithm’s ultimate goal is to mimic the ophthalmologists’ decisions and provide the ability to analyse large volumes of data with high accuracy.

The ground truth for the convolutional neural network (CNN)-based CNV boundary segmentation and the peripheral arcades/dark halo features were manually delineated by one experienced retina specialist (EKP) and double-checked by another (HRE). These two retina specialists (EKP and HRE) visually examined all of the en-face images to provide labels for the presence/absence of each CNV activity criteria feature, including the shape, branch, peripheral arcade, anastomosis and loops, and dark halo. All images were labelled independently by the experts, and the standard was confirmed if there was no difference for any specific image. In addition, the ophthalmologists engaged in debate and adjudication until complete consensus was established. The presence or absence of the feature is determined by ophthalmologists using the outer retina en-face images for the first four features (shape, branch, peripheral arcade, anastomosis, and loops); however, the choroidal en-faces are the reference for the last feature (dark halo). The demographic characteristics of the dataset are shown in Table 1.

The block diagram of the CNV-Net is illustrated in Figure 2. The algorithm has two main stages, including a segmentation block and a binary classification block, each of which are elaborated in the subsequent sections. 

### 2.2. Segmentation Algorithm

#### 2.2.1. Segmentation of CNV Area (SEG-CNV)

The goal of this segmentation task is to identify the precise location of the CNV lesion in the outer retinal en-face images. We modified the U-net algorithm, which is a well-known model for segmenting medical images and includes encoder and decoder routes [17]. Figure 3 depicts our SEG-CNV architecture, which is based on a redesign of the original U-net. In the original format, approximately 30 million parameters must be trained, resulting in overfitting in small datasets. We reduced the number of parameters in SEG-CNV to about 1.6 million for grayscale input images of 128 × 128 pixels. The white pixel on a black background represents the output CNV lesion (CNV mask in Figure 3). As precise information from the internal structure of the CNV is required in the classification stage, the predicted mask is multiplied by the outer retinal en-face image. This produces a CNV region of interest (ROI) as the segmentation’s final output (CNV ROI in Figure 4).

The Dice loss function was computed by calculating the overlap between the segmented area (CNV) and the ground-truth (GT), as follows:(1)$$L=1-\frac{2\times (CNV\cap GT)}{CNV\cup GT}$$

#### 2.2.2. Segmentation of Peripheral Arcade (SEG-PA)

Peripheral arcades are small anastomotic and looping vessels that branch into the vascular arcades between the vessel termini. Accordingly, a specific algorithm, called SEG-PA (Figure 4), is designed to extract the location of such loops in the outer retinal enface images. First, the vessels are identified and binarised such that white pixels represent vessels (first image in the SEG-PA box of Figure 4). By inverting the color within the CNV mask, the spaces between the vessels are made white (second image in the SEG-PA box). Finally, to concentrate on the location of the peripheral arcade, this image is multiplied by a ring-shaped mask (calculated by applying dilation on the CNV mask), to retain the spaces between the vessels at the edge of the CNV region. This is how the peripheral arcade mask is finally obtained (Figure 2 and Figure 4). 

#### 2.2.3. Segmentation of Dark Halo (SEG-DH)

The SEG-DH algorithm was designed to segment the dark region around the CNV mask in the choriocapillaris en-face images to aid in the detection of the halo activity criteria. The region-growing algorithm does this by assembling similar pixels into an initial region [18]. We segmented the CNV mask on the outer retina en-face using the SEG-CNV algorithm to define an initial region based on the anatomical location of the dark halo surrounding the CNV area. The CNV mask boundary location was then copied to an enhanced version of the choroidal en-face image and sent to the region-growing algorithm. Finally, as shown in Figure 4, the dark halo mask is obtained.

### 2.3. Clasification Algorithm

The classification task aims to categorise the activity of CNV lesions based on five morphological features (branch, peripheral arcade, shape, anastomosis and loops, and dark halo). The above-mentioned segmentation block provides the input to this algorithm. Each feature is assigned a label (1 for presence and 0 for absence) by five individual classifiers. Multiple binary networks have been shown to improve the classification accuracy, classification speed, and resource consumption [19].

Deep networks have a large number of trainable parameters, which allow them to generalise well. However, a large amount of labelled data is required to prevent over-fitting. Transfer learning (TL), on the other hand, is an effective method for addressing this issue, as it is based on the premise that the characteristics extracted by a pre-trained model can be reused in other categorisation tasks [20,21,22]. As the dataset is small (limited to 130 cases), four out of five classifiers benefit from TL using previously trained models [23]. We used the VGG16 framework for TL, which was initially trained with ImageNet data [24]. VGG16’s architecture, which includes convolutional and pooling layers, fully connected (FC) layers, and SoftMax, is used in the design of the classifiers for the branch, shape, anastomosis and loops, and dark halo features. Except for a few convolutional layers and the output FC layer, the VGG16 weights are kept frozen (elaborated in Table 2). As a result, the number of trainable parameters is reduced from 138 million to around 641,000, lowering the risk of overfitting the TL. Despite the fact that the majority of the parameters are learned from ImageNet, the performance is good for unrelated CNV images. The reasoning behind this performance is that the learned ImageNet parameters are related to the primary layers dedicated to the general image specification (common between ImageNet and CNV); however, specific features of the new CNV dataset are learned by the parameters trained after TL. The only activity criteria classified without TL were the peripheral arcade features, for which a small DL network built from scratch was chosen due to its superior performance when compared to TL methods. (Details are provided in Table 2). Similar to the segmentation section, data augmentation (such as rotating, zooming, flipping, and shearing) was used to expand the training data. To reduce the computational complexity, the input images were resized to 224 × 224 × 3 for the TL using VGG. Nested five-fold cross-validation is used due to the limited number of images in the dataset and to provide a more reliable evaluation. The folds are divided subject-by-subject to ensure that images from the same patient do not appear in both the training and testing data (yielding unwanted information leakage).

## 3. Results

In this section, we present the experiments that were carried out, as well as the results of the segmentation and classification blocks.

### 3.1. Metrics for Segmentation and Classification

The segmentation task is evaluated using overlap-based metrics. The true positives (TP), false positives (FP), true negatives (TN), and false negatives (FN) cardinalities of the so-called confusion matrix are then used to evaluate the classification task.

Different metrics are used to assess the success rate of the segmentation techniques. Accuracy is expressed as a percentage of the correctly segmented pixels in an image. The Dice similarity coefficient is another metric for evaluating the performance in terms of the normalised overlap between the ground truth and predicted mask (Equation (1)) [25]. The Dice values range between 0 and 1, with 0 indicating no spatial overlap and 1 indicating complete overlap.
(2)$$Dice=\frac{2\times \left|CNV\cap GT\right|}{\left|CNV\right|+\left|GT\right|}=\frac{2TP}{2TP+FP+FN}$$

To evaluate the performance of the classification method, the accuracy, specificity, F1 score, and sensitivity were used (Equations (3)–(6)).
(3)$$Accuracy=\frac{TP+TN}{TP+FP+FN+TN}$$
(4)$$F1\_score=\frac{2TP}{2TP+FP+FN}$$
(5)$$Specificity=\frac{TN}{TN+FP}$$
(6)$$Sensitivity=\frac{TP}{TP+FN}$$

#### 3.1.1. Result for Segmentation

Three segmentation algorithms are proposed in this work, and this section is devoted to analysing the performance of each algorithm, individually. The customized U-net performance for the segmentation of the CNV area (SEG-CNV) and the comparison with the original U-net is demonstrated for the segmentation of the CNV area; the Dice coefficient was 0.90 with our customised U-net SEG-CNV, up from just 0.62 with the original U-net [21,23]. Our proposed segmentation network yielded a mean Dice coefficient of 0.86, achieving at least 0.69 on any individual image. Figure 5 illustrates the distribution of the Dice coefficients over the image set. Examples of the good performance of the SEG-CNV algorithm (CNV-masks and CNV-ROI, as defined in Figure 2) are demonstrated in Figure 6.

SEG-PA is designed to localise the peripheral arcade, as depicted in Figure 7. Figure 7B,F shows an area to check for the presence or absence of the peripheral arcade features in the CNV lesion. To show the performance of the SEG-DH algorithms, Figure 8 shows the original choroidal enface image and the area predicted by SEG-DH.

#### 3.1.2. Result for Classification

Five individual binary classification algorithms are developed to assign a label (0 or 1) to each of the five morphological features (branch, shape, anastomosis and loop, peripheral arcade and dark halo). Details of each algorithm and the corresponding input are provided in Table 1. Four models are developed with TL and a small DL network is designed from scratch.

Regarding the TL models, the best number of trainable layers is determined to yield the best performance. The model encounters overfitting when a high number of layers are trained using the limited available data.

The results of each individual binary classification are presented in Table 3, using the optimal hyper parameters found with a grid search. Figure 9 demonstrates the confusion matrices for each classifier, providing a visual representation of the agreement between the ground truth labels and the predictions of the proposed approach.

### 3.2. Model Comparison

We found that the results of the TL-based algorithm (VGG16) with the usage of the SEG-PA and SEG-DH technique were superior to those of the DL-based models without the use of these methods. For each binary classification, the results are shown in Table 4, using deep learning instead of TL.

## 4. Discussion

The current study aimed to develop an AI-based segmentation and classification algorithm for the evaluation of CNV activity based on the five activity features in OCTA images originally introduced by Cascos and colleagues. The proposed algorithm is divided into two steps. The first block involves using a modified U-Net network to pre-process and segment CNVs in OCTA images. The second block consists of five binary classification networks, implemented with DL models, and using transfer learning from pre-trained networks. The proposed segmentation network yielded an averaged Dice coefficient of 0.90 across all of the images, with a performance in any individual image no worse than 0.61 in any image. The individual classifiers corresponding to five activity criteria (branch, peripheral arcade, dark halo, shape, loop and anastomoses) showed accuracies of 0.84, 0.81, 0.86, 0.85, and 0.82, respectively. 

Previous AI studies in nAMD have demonstrated a satisfactory ability to predict the conversion of intermediate-type AMD to nAMD in the same eye and the fellow eye, the recognition of predictive biomarkers for AMD progression, and the study of fluid distribution during anti-VEGF therapy for nAMD [26,27]. OCTA en-face images are relatively new in the context of image processing, and just a few studies have been presented on their automatic analysis. Liu et al. [28] developed an automated saliency technique for CNV region detection. To overcome problems such as projection artefacts and CNV diversity in outer retinal en-face images, their algorithm employed noise cancellation and a conspicuous detection approach. On the basis of the Jaccard (similarity metric) criteria, quantitative and qualitative analyses of scans from seven participants revealed an accuracy rate of 0.83. In a later study involving many of the same authors, Xue et al. [29] proposed a CNV segmentation technique based on an unsupervised and parallel machine learning technique named density cell-like P systems. On 22 pictures with neovascular AMD, the model was judged to be 87%accurate. When noise pixels were bright and near a new blood vessel, their approach was unable to correctly separate the vascular information from the noise. In a study conducted by Taibouni et al. [30], to quantify neovascular AMD biomarkers, two algorithms were created for the CNV analysis of two distinct groups of OCTA en-face images (Group 1 consisted of tightly packed, high-flow networks with no discernible branching. Group 2 exhibited a neovascular network with distinct branching patterns and substantial blood flow). Each algorithm involved pre-processing, segmentation, and quantification. In Group 2, however, where individual branching of the neovascular network was evident, en-face image analysis was added to the additional vascular enhancement phase in order to maintain the vascular pattern’s finer characteristics. They tested the algorithm using 54 OCTA outer retinal en-face images from 54 patients (24 images in group 1 and 30 images in group 2) and reported an 87% accuracy using the Jaccard criterion. 

Due to the extensive use of Deep Learning (DL) algorithms in ocular image analysis, a number of recent DL-based studies were customised to analyse OCTA images [31]. In a recent study, Wang et al. [32] used 1676 outer retina en-face images (with and without CNV) to run a DL-based algorithm to identify and segment CNV. Their approach was based on two convolutional neural networks (CNN), the first to classify the presence of CNV and the second to segment the vessels in CNV. They achieved an F1 score of 0.93. 

Recently, Jin et al. [31] proposed a multimodal deep learning (DL) model using OCT and OCTA images for the assessment of CNV activity in nAMD. They used a novel feature-level fusion method to combine the multimodal data and showed an accuracy of 95.5% to detect active CNV. However, they assessed the CNV activity based on the presence of fluid and the CNV activity criteria in OCT and OCTA images, respectively. In the OCTA images, they did not evaluate each CNV feature independently. 

To the best of our knowledge, this is the first attempt to automatically calculate the activity criteria for CNV lesions in OCTA images. We proposed a two-step technique for segmenting the CNV mask and assigning activity criteria to each lesion. For this reason, en-face images of the outer retina and choroid were employed. During the segmentation phase, the DL technique was used to define the region of the outer retina containing the CNV (CNV mask). By multiplying the CNV mask with the original image, we obtained a CNV ROI, in which the undesired noisy surrounding area was suppressed, while the background of the vessels, which is required for the subsequent feature extraction, was retained. In addition, the dark halo mask and peripheral arcade mask were segmented from the outer retina and choroidal en-face images, respectively, using conventional image processing techniques. These three masks are then sent into a DL-based classification step where the five activity criteria were automatically assigned to each CNV lesion. To this end, each activity criterion’s existence or absence was determined by an individual binary DL classification model. Transfer learning (TL) techniques utilising pre-trained models were applied for all features, except peripheral arcade, due to a scarcity of data. For the peripheral arcade features, a small DL network built from scratch was chosen due to its superior performance when compared to TL methods. The network was then customised by re-training a small number of network parameters using the data supplied. The ultimate objective of the proposed CNV-Net method was to simulate the decisions made by ophthalmologists and to enable the accurate analysis of vast volumes of data.

This AI-based algorithm potentially allows the reliable detection and segmentation of CNV from OCTA alone, without the need for imaging with contrast agents. The evaluation of the activity criteria in CNV lesions obtained acceptable results, and this algorithm could enable the objective, repeatable assessment of the CNV features.

We acknowledge that our research has several limitations, which serve to stimulate and support future projects. The research is limited by the small sample size and cross-sectional design. In order to achieve more comprehensive results in the future, we would need to enlarge the training dataset or employ other data augmentation techniques. In the classification section of the proposed method, the presence or absence of each activity criterion is determined by five binary classification DL models. We intend to evaluate our CNV-Net approach with a multi-classification algorithm in the future. The algorithm’s individual decisions might be supervised by saliency map volume scan visualisation. Even though the saliency map focuses on clinically significant locations, it should be interpreted with caution due to the small dataset in comparison to the variety of vascular patterns. To successfully validate our method for the diagnosis and treatment of nAMD, further CNV lesion analyses on OCTA images must be performed.

## 5. Conclusions

CNV is a vision-threatening development in a variety of common retinal diseases. The current clinical gold standard involves dye-based imaging, but this is invasive, burdensome for patients, and is relatively costly. Thus, there is excitement around the possibility of detecting and monitoring CNV non-invasively via OCTA, with associated benefits for patients and the public. In clinical research using OCTA images, CNV regions are frequently drawn or segmented by manually modified thresholding, which is time-consuming, especially as the influence of the image quality and artefacts must be carefully considered for the accurate classification of lesions. In this paper, we showed a novel CNV segmentation and classification approach utilizing deep learning on OCTA images. Two objectives were accomplished in this work: first, the algorithm correctly segmented CNV regions in the outer retina and choroidal en-face; second, the algorithm successfully classified various features of CNV lesions as absent or present. This algorithm could be potentially used in the future for improved disease monitoring and treatment. 

The most commonly used metrics for semantic segmentation is the Dice Coefficient. We also added other typical metrics used with segmentation problems (Sensitivity and the Precision). The results are also presented with the K-folds cross validation technique, a popular method that results in less biased outputs. It ensures that every observation from the original dataset has the chance of appearing in the training and test set and is a good substitute for the Train/Test split when there is a limited set of input data.

To compare the proposed methods and discuss the advantages and disadvantages of having different parts of the algorithm, there is no doubt that an algorithm consisting of two successive stages of segmentation and classification to solve the image processing challenge is very complicated and requires a relatively high processing time. Nevertheless, in this research, we accepted the complexity of the semantic segmentation algorithm to achieve higher accuracy in the classification task. In other words, the segmentation algorithm separates the region of interest, the CNV lesion, from the unnecessary information in the background of the OCTA images. In this way, the semantic segmentation algorithm helps to improve the understanding of the target by the classification algorithm. However, when a classification algorithm is used alone, although the time to perform the algorithm is shortened, the network pays attention to the distractive features and unnecessary information in the image’s background. The use of a single-stage algorithm that only includes a model with a classification task yields less computational complexity, but due to a series of distractors in the input data, there is an increase in false negatives. In addition, data augmentation and transfer learning are applied to overcome the lack of the tagged OCTA dataset. Furthermore, by comparing Table 3 and Table 4 reported in the manuscript, the effect of using two segmentation methods based on classical methods clearly shows the improvement in the accuracy of the classification algorithm.

The AI-based algorithm potentially allows the reliable detection and segmentation of CNV from OCTA alone, without the need for imaging with contrast agents. The evaluation of the activity criteria in CNV lesions obtained acceptable results, and this algorithm could enable the objective, repeatable assessment of CNV features.

## Figures and Tables

**Figure 1 diagnostics-13-01309-f001:**
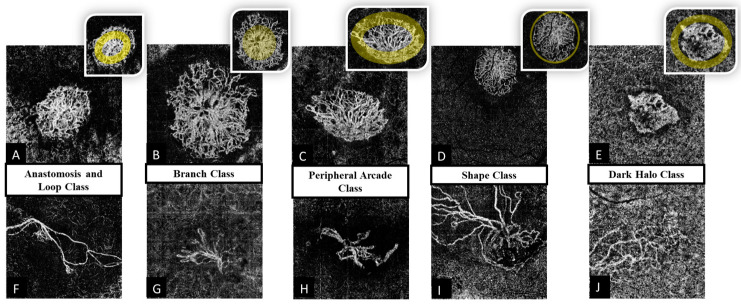
Example of presence (upper row) or absence (lower row: (**F**–**J**)) of five features of CNV activity in ten separate patients. (**A**) anastomosis and loops, (**B**) branching, (**C**) peripheral arcade, (**D**) shape, (**E**) dark halo. Insets show the highlighted region of CNV that define the feature.

**Figure 2 diagnostics-13-01309-f002:**
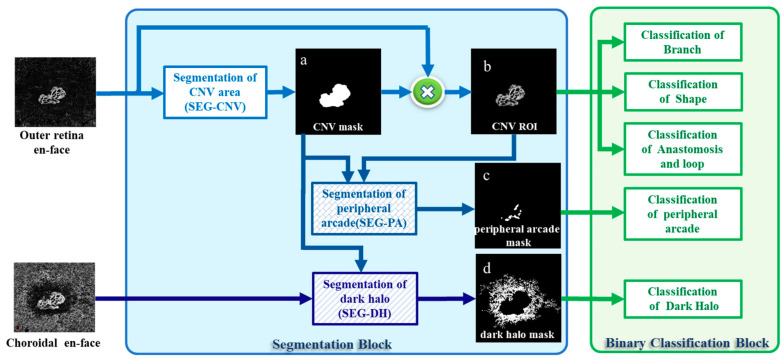
Block diagram of the CNV-Net method including segmentation and classification blocks. (**a**) Output of SEG-CNV algorithm (CNV Mask), (**b**) Multiplication of CNV mask and the outer retina en-face image (CNV ROI), (**c**) Output of SEG-PA algorithm (peripheral arcade mask), (**d**) Output of SEG-DH algorithm (dark Halo Mask).

**Figure 3 diagnostics-13-01309-f003:**
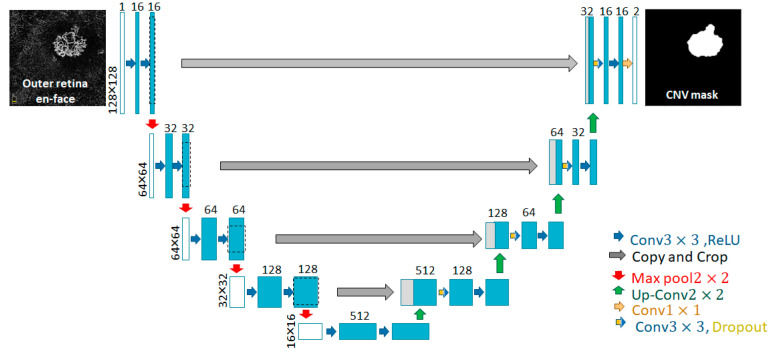
Structure of SEG-CNV (customised U-net network). The red, orange, green and blue arrows represent transfer function, convolution function with 1 × 1 convolution kernel and sigmoid function, up-sampling function, convolution function with 3 × 3 convolution kernel and RELU function, respectively. Blue/yellow arrows indicate convolution with dropout.

**Figure 4 diagnostics-13-01309-f004:**
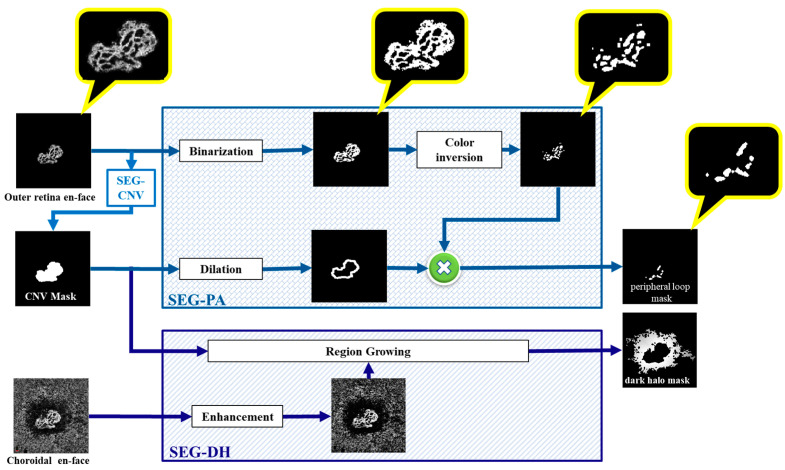
Proposed algorithm for SEG-PA and SEG-DH.

**Figure 5 diagnostics-13-01309-f005:**
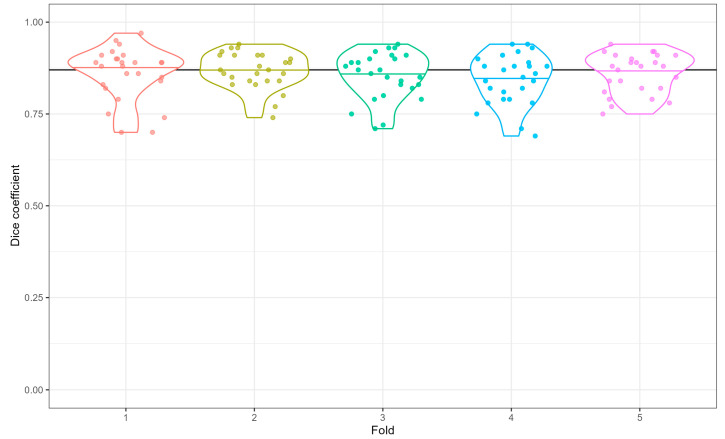
Distribution of Dice coefficients, measuring the performance of our CNV segmentation algorithm, each point represents the Dice coefficient for one image in a particular fold. The violins show the distribution. Coloured horizontal lines mark the median D Dice coefficient within one-fold; the black line is the median overall.

**Figure 6 diagnostics-13-01309-f006:**
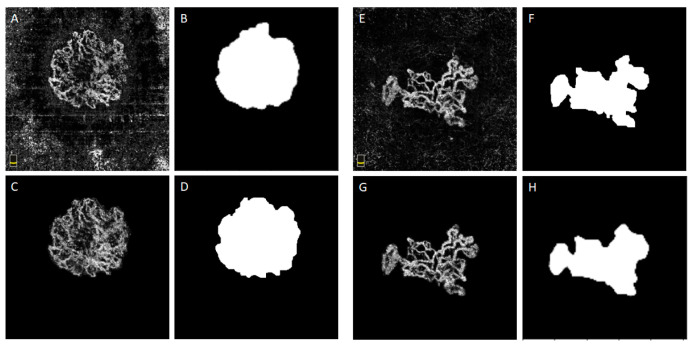
Examples of performance of SEG-CNV (**A**,**E**) in outer retinal en-face images. (**B**,**F**) Ground Truth, (**C**,**G**) Predicted CNV mask by SEG-CNV represented by the white pixel on black background, (**D**,**H**) CNV ROI by SEG-CNV as final output of this segmentation.

**Figure 7 diagnostics-13-01309-f007:**
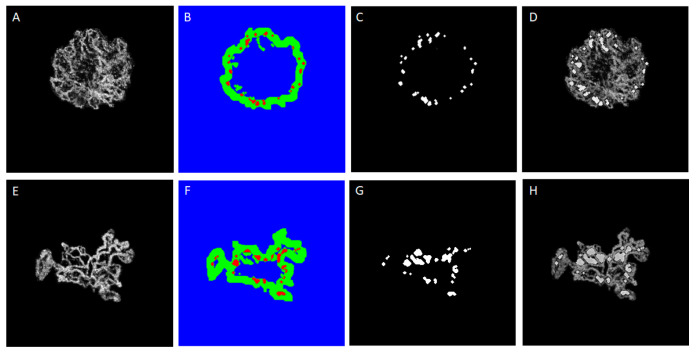
Examples of the performance of SEG-PA to localise peripheral arcade on CNV lesion (**A**,**E**) CNV-ROI, (**B**,**F**) localised peripheral arcade (red points) and the peripheral area (green), (**C**,**G**) localised peripheral arcade overlaid on CNV-ROI, (**D**,**H**) peripheral loop mask as the main output of this algorithm.

**Figure 8 diagnostics-13-01309-f008:**
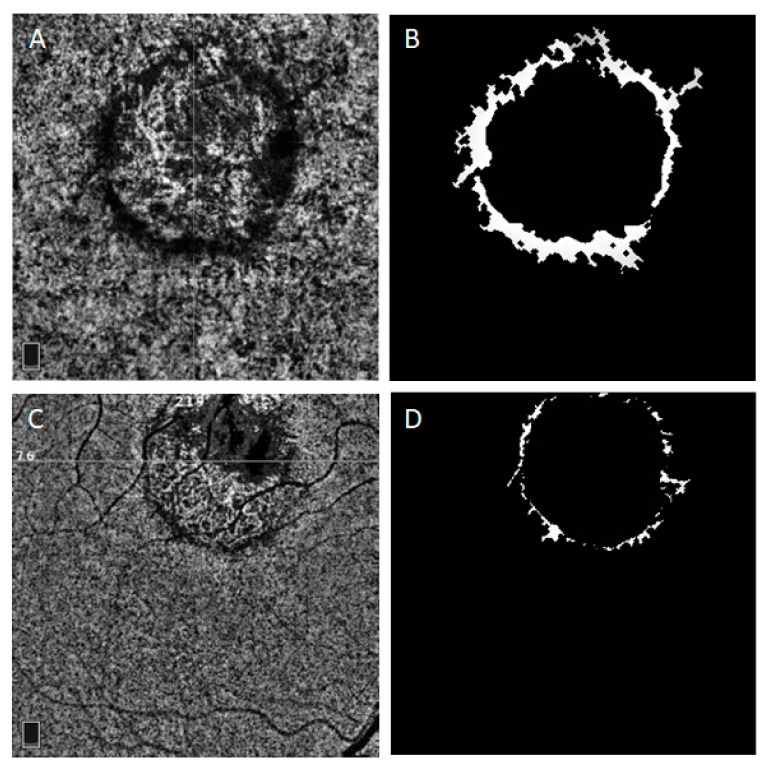
Result of SEG-DH. (**A**,**C**) original choroidal enface, (**B**,**D**) the predicted area by CNV-DH.

**Figure 9 diagnostics-13-01309-f009:**
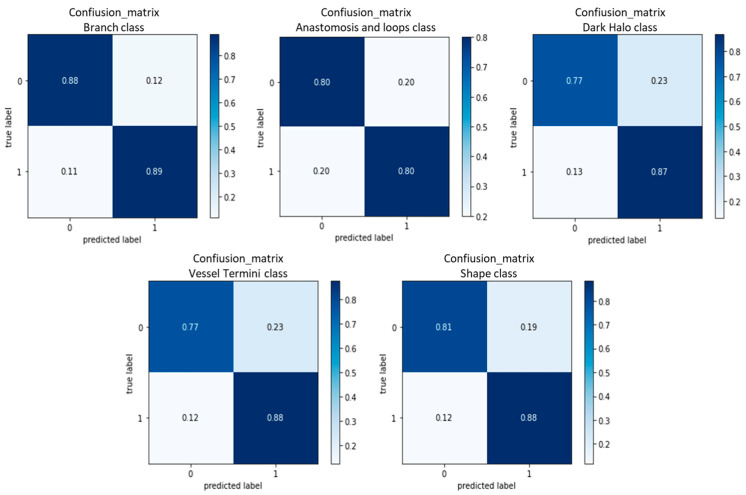
Representative confusion matrices of 5-category binary classification (Table 1 shows the number of images falling into each category). These matrices show the average performance across different folds.

**Table 1 diagnostics-13-01309-t001:** Demographic characteristics of the patients in the dataset.

Variable	Values	Number (Percentage)
Sex	Female	52 (40%)
	Male	78 (60%)
Total		130 (100%)
Age	<40	9 (6.92%)
	40–50	14 (10.77%)
	50–60	25 (19.23%)
	60–70	37 (28.46%)
	70–80	45 (34.61%)
Total		130 (100%)
Outer retina en-face Total = 130		Presence (Percentage)
	branch	68/130 (52.30%)
	shape	49/130 (37.69%)
	peripheral arcade	100/130 (76.92%)
	anastomosis and loops	63/130 (48.46%)
Choriocapillaris en-face Total = 130	Dark halo	62/130 (47.69%)

**Table 2 diagnostics-13-01309-t002:** Description of CNV lesion classification models.

Input En-Face Image	Segmentation Block		Classification Block		Activity Criteria
	Segmentation Method	Output Image	Method		
			DL Model	Trained Layers	
Outer retina	SEG-CNV	CNV ROI	TL on VGG16	FC + Sigmoid	Branch
Outer retina	SEG-CNV	CNV ROI	TL on VGG16	3Conv + FC + Sigmoid	Shape
Outer retina	SEG-CNV	CNV ROI	TL on VGG16	1Conv + FC + Sigmoid	Anastomosis and Loops
Outer retina	SEG -PA	Peripheral Arcade Mask	DL from scratch	All layers (4Conv + FC + Sigmoid)	Peripheral Arcade
Choroidal	SEG-DH	Dark Halo Mask	TL on VGG16	FC + Sigmoid	Dark Halo

**Table 3 diagnostics-13-01309-t003:** Classification metrics for five classifier models.

Features Metrics	Branch	Shape	Anastomosis and Loops	Peripheral Arcade	Dark Halo
**F1-score**	0.83	0.86	0.79	0.82	0.83
**Sensitivity**	0.86	0.86	0.83	0.86	0.81
**Specificity**	0.70	0.81	0.76	0.78	0.83
**Accuracy**	0.84	0.85	0.82	0.81	0.86

**Table 4 diagnostics-13-01309-t004:** Classification metrics for five classifiers without the use of the SEG-PA and SEG-DH method.

Features Metrics	Branch	Shape	Anastomosis and Loops	Peripheral Arcade	Dark Halo
**F1-score**	0.79	0.79	0.65	0.78	0.68
**Sensitivity**	0.75	0.70	0.60	0.71	0.59
**Specificity**	0.72	0.79	0.78	0.77	0.67
**Accuracy**	0.60	0.74	0.70	0.79	0.65

## Data Availability

The codes are freely available in: https://github.com/mahsavali/CNV-Segmentation-Classification-OCTA (accessed on 23 March 2023).

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
