# Peer review of "CNV-Net: Segmentation, Classification and Activity Score Measurement of Choroidal Neovascularization (CNV) Using Optical Coherence Tomography Angiography (OCTA)"

_diagnostics, 2023, doi:10.3390/diagnostics13071309_

Round 1

Reviewer 1 Report

The authors proposed a CNV-Net model for the automatic segmentation of choroidal neovascularization from angiography. The manuscript is well-written and it has scientific sounds. However, my concern is an evaluation measure, more specific evaluation indexes of the image segmentation should be used for automatic segmentation tasks using the proposed method. 

Author Response

Thanks for the useful comments, the answer to reviewers is atttached.

Reviewer 2 Report

This is a good research work which is useful for readers. Some issues may be address to improve this paper:

(1) Why (Coscas et al.2015) reference is used in the citation?

(2) One comments arised are that there is no enough discussion to compare methods in terms of their advantages and disadvantages (please provide them in a supplementary file) 

(3) Conclusion Section is too short. Can you explain in details.

(4) Optional: Can authors able to provide codes or software publicly for readers help?

Author Response

(The authors gave the same response as above.)
